# A Deep-Sequencing Workflow for the Fast and Efficient Generation of High-Quality African Swine Fever Virus Whole-Genome Sequences

**DOI:** 10.3390/v11090846

**Published:** 2019-09-11

**Authors:** Jan H. Forth, Leonie F. Forth, Jacqueline King, Oxana Groza, Alexandra Hübner, Ann Sofie Olesen, Dirk Höper, Linda K. Dixon, Christopher L. Netherton, Thomas Bruun Rasmussen, Sandra Blome, Anne Pohlmann, Martin Beer

**Affiliations:** 1Friedrich-Loeffler-Institut, Federal Research Institute for Animal Health, Südufer 10, 17493 Greifswald - Insel Riems, Germany; 2Animal Health Diagnostic Laboratory, Republican Center for Veterinary Diagnostic, str. Murelor, 3, MD-2051, mun. Chișinău, Moldova; 3Department of Veterinary and Animal Sciences, Faculty of Health and Medical Sciences, University of Copenhagen, DK-1870 Frederiksberg C, Denmark; 4The Pirbright Institute, Ash Road, Pirbright, Woking GU24 0NF, UK; 5DTU National Veterinary Institute, Technical University of Denmark, Lindholm, DK-4771 Kalvehave, Denmark; 6Department of Virus & Microbiological Special Diagnostics, Statens Serum Institut, DK-2300 Copenhagen S, Denmark

**Keywords:** African swine fever virus (ASFV), next-generation sequencing (NGS), whole-genome sequencing, Nanopore sequencing, target enrichment

## Abstract

African swine fever (ASF) is a severe disease of suids caused by African swine fever virus (ASFV). Its dsDNA genome (170–194 kbp) is scattered with homopolymers and repeats as well as inverted-terminal-repeats (ITR), which hamper whole-genome sequencing. To date, only a few genome sequences have been published and only for some are data on sequence quality available enabling in-depth investigations. Especially in Europe and Asia, where ASFV has continuously spread since its introduction into Georgia in 2007, a very low genetic variability of the circulating ASFV-strains was reported. Therefore, only whole-genome sequences can serve as a basis for detailed virus comparisons. Here, we report an effective workflow, combining target enrichment, Illumina and Nanopore sequencing for ASFV whole-genome sequencing. Following this approach, we generated an improved high-quality ASFV Georgia 2007/1 whole-genome sequence leading to the correction of 71 sequencing errors and the addition of 956 and 231 bp at the respective ITRs. This genome, derived from the primary outbreak in 2007, can now serve as a reference for future whole-genome analyses of related ASFV strains and molecular approaches. Using both workflow and the reference genome, we generated the first ASFV-whole-genome sequence from Moldova, expanding the sequence knowledge from Eastern Europe.

## 1. Introduction

Over the last decade, tremendous progress has been made in the field of DNA sequencing [1]. With the introduction of modern next-generation sequencing (NGS) platforms—e.g., Roche 454, Illumina, Ion Torrent and PacBio—the amount of generated sequences has dramatically increased while the costs have rapidly decreased [1,2]. Especially in the field of emerging infectious diseases, whole-genome sequencing using NGS has become an essential and widespread tool in understanding molecular epidemiology and pathogen evolution [3,4].

One emerging animal pathogen for which the need for whole-genome sequences has massively increased in recent years is African swine fever virus (ASFV) [5].

ASFV was first described and is endemic in sub-Saharan Africa [6] where it can be detected in wild African suids—e.g., forest hog, red river hog and bushpigs—and is transmitted in a sylvatic cycle between warthogs and *Ornithodoros* soft ticks [7]. Since only distantly related giant viruses with double-stranded DNA genomes are known so far, ASFV is the only member of its genus *Asfivirus* and family *Asfarviridae* [5]. ASFV causes a disease in domestic pigs and European wild boar called African swine fever (ASF), which is characterised by high fever, haemorrhages, respiratory, gastro-intestinal and neurological signs and cyanosis [8,9]. The animals show a lethality of up to 100% depending on the strain-specific virulence and usually succumb to the infection within 7 to 10 days after infection [9]. Neither vaccines nor treatments are available [10]; thus, stopping ASFV spread relies solely on strict quarantine measures including the culling of infected herds, establishment of restriction zones and movement control leading to enormous socio-economic consequences in affected countries [11,12].

From Africa, ASFV was introduced into Europe (Portugal) two times and subsequently spread to other European countries; e.g., Spain, France, Italy, Belgium and the Netherlands [13,14]. After the virus was eradicated from most affected countries (with the exception of Sardinia, where ASFV is still endemic) [11], ASFV was reintroduced in Georgia in 2007 [15]. Since then, the virus has subsequently spread through wild boar and domestic pig populations of Eastern Europe (Georgia, Armenia, Russia, Belarus, Moldova, Ukraine) [13,16,17,18] and the European Union (Bulgaria, Hungary, Czech Republic, Slovakia, Romania, Estonia, Latvia, Lithuania and Poland) [18]. More recently, the virus spread to Western Europe (Belgium) [19] and reached Asia, causing devastating outbreaks (China, Cambodia, Mongolia, Vietnam, North Korea, Laos and Myanmar) [13,20].

The sequencing of partial genomes by Sanger sequencing has been used for decades to identify viral genotypes and trace virus introductions through molecular epidemiology, especially in Africa where ASFV is widely distributed and different genotypes occur [21,22,23,24,25,26,27]. The viral genome consists of one linear molecule of double-stranded DNA with a length of 170–194 kbp [28]. It shows a low mutation rate typical for DNA viruses that leads to a very low genetic variability, especially in close geographic regions [23,29,30]. However, the identification of attenuated viruses with large deletions, inversions or duplications as well as novel genotypes in Africa might be explained by mechanisms leading to genetic reorganisations such as homologous recombination, as was suggested and described for other large dsDNA viruses [31]. Therefore, the analysis of partial sequences quickly becomes ineffective for detailed analyses of phylogeny, molecular epidemiology, or virus evolution, as observed especially in Eastern Europe [19].

The first ASFV whole-genome sequence was published in 1995 and generated from the Spanish cell-culture-adapted ASFV strain BA71V using Sanger sequencing [32,33]. Since then, only a few additional genomes have been published using the same technique [34,35] (Table 1). In 2009, the first ASFV genome sequence was published using the Roche 454 GS FLX NGS technique in combination with Sanger sequencing [36]. Subsequently, different platforms have been used to generate ASFV whole-genome sequences including Illumina HiScanSQ, MiSeq, HiSeq, NextSeq500 and PacBio [37,38,39,40,41,42,43,44,45,46] (Table 1).

Since the ASFV genome includes extensive homopolymer and repeat regions including inverted terminal repeats (ITR) of variable length [28], platform-specific limitations especially in homopolymer sequencing need to be considered [47,48]. In particular, the short read data generated by sequencing platforms of the second generation—e.g., Roche 454, Illumina and IonTorrent—need to be handled with care. Used for mapping against a reference sequence, genomic reorganisations such as inversions or duplications might be missed, and the quality of the consensus-sequence strongly relies on the sequence used as reference. Furthermore, when used for assembly, the small reads (200–300 bp) might have only small overlapping regions and therefore can be misassembled and—especially in homopolymer and repeat regions—the quality of the consensus-sequence might be low.

Therefore, data on sequence quality such as coverage, mappings and alignments are crucial to assess the results of whole genome sequencing and the analysis of genetic variations including single nucleotide variants (SNV) analyses and molecular epidemiology experiments [3].

In particular, reference sequences frequently used as a basis for experiments and analyses [45,49,50] need careful validation. Especially for ASFV, where only little data on gene expression and protein translation are available and most of the between 150 and 167 ORFs have only been predicted [28,37], reliable whole genome sequences are the basis needed for transcriptome and proteome analyses and up-to-date annotations.

Here, we describe a sequencing protocol combining ASFV-specific target enrichment, Illumina sequencing and long-read Nanopore sequencing to generate high-quality complete ASFV genome sequences. We used the described protocol for resequencing of the ASFV Georgia 2007/1 genome, leading to the correction of 71 homopolymer errors and other sequencing artefacts, a longer ITR region and ultimately updated annotations [37] of the corrected genome sequence. Using this high-quality sequence as a reference, we used target enrichment in combination with Illumina sequencing and sequenced ASFV directly from organ tissue, thereby providing an ASFV whole-genome sequence from an outbreak in Moldova.

## 2. Materials and Methods

### 2.1. Virus Cultivation

The original ASFV Georgia 2007/1 isolate from Georgia reported in 2011 [37] was passaged once on porcine bone marrow cells (PBMs) prior to sequencing.

To generate high molecular weight (HMW)-DNA to facilitate long reads through Nanopore sequencing, ASFV Georgia 2007/1 was passaged a second time on porcine primary peripheral blood monocytes (PBMCs) as described elsewhere [54].

### 2.2. DNA Extraction

For Illumina sequencing, DNA was extracted from PBMC cell culture supernatant using the NucleoMag® VET Kit (Macherey Nagel, Düren, Germany) according to the manufacturer’s instructions.

The Moldovan organ sample (spleen), collected from infected domestic pigs, was homogenised in a 2 mL reaction tube containing one 5 mm steel bead and 200 µL phosphate buffered saline (PBS) in a TissueLyser II (Qiagen, Hilden, Germany) for 3 min at 30 Hz. After centrifugation at 10,000× *g* for 3 min, DNA was extracted from the supernatant as described above.

For Nanopore sequencing, HMW-DNA was extracted using a customised protocol. PBMCs [54] were infected with ASFV Georgia 2007/1 at an MOI of 0.1 and harvested after 24 h. After one freeze-thaw-cycle of infected PBMCs, cell debris was removed by centrifugation at 1000× *g* for 10 min at 4 °C and the virus particles were concentrated using ultracentrifugation through a sucrose cushion. Briefly, 6 mL 40% sucrose (in PBS) were carefully overlaid with the virus-containing supernatant (30 mL) and centrifuged for one hour at 50,000× *g* at 4 °C. The resulting pellet was resuspended in 100 µL PBS and heat-inactivated for 30 min at 75 °C. Subsequently, the virus suspension was mixed with 3 mL TEN buffer (40 mM Tris-HCl, 1 mM EDTA pH 8.0, 150 mM NaCl) and 1.5 mL sarcosyl buffer (3% sodium lauroyl sarcosinate, 75 mM Tris-HCl pH 8.0, 25 mM EDTA) and incubated at 65 °C for 15 min. The DNA was extracted using equal volumes of phenol (Roth) in a first step, phenol/chloroform/isoamyl alcohol (Roth) in a second step and finally chloroform (Roth)/isoamyl alcohol (Merck) (24:1) in a final step. Precipitation of the DNA was carried out using 0.1 volumes of sodium acetate buffer (3 M, pH 4.8) and 2.5 volumes of ethanol at −20 °C for 16 h. The precipitated DNA was pelleted and washed with 70% ethanol and air-dried before final resuspension in 70 µL EB buffer (Qiagen).

### 2.3. Illumina Sequencing

Samples were prepared for and analysed on an Illumina MiSeq platform in the 300 bp paired-end mode, as previously described [55].

### 2.4. Nanopore Sequencing

Library preparation was carried out using the Rapid Barcoding Sequencing Kit (SQK-RBK004, Oxford Nanopore Technologies, Oxford, UK) according to the manufacturer’s instructions. The finished library was loaded onto a R9.5.1 MinION flow cell (FLO-MIN107, Oxford Nanopore Technologies). Sequencing was conducted under the standard settings using a MinION (Mk1B, Oxford Nanopore Technologies) in combination with a MinIT (MNT-001, Oxford Nanopore Technologies). Signal information (fast5) was basecalled with MinIT and Guppy v2.1.3 (Oxford Nanopore Technologies).

### 2.5. Target Enrichment

For the enrichment of ASFV-specific sequences prior to Illumina MiSeq sequencing, the myBaits® kit with 1–20K unique baits (Arbor Bioscience, Ann Arbor, MI, USA) was used according to the manufacturer’s instructions (Figure 1). Briefly, RNA baits (11,958 unique baits) were designed in silico to cover the ASFV Georgia 2007/1 sequence (FR682468.1) and the ASFV Ken06.bus sequence (KM111295) with a three-fold coverage by Arbor Bioscience. After DNA extraction and standard library preparation [55], 100–500 ng of indexed libraries were heat-denatured at 95 °C and, after cooling to hybridisation temperature (65 °C), adapter sequences were blocked using Illumina-specific blocking oligos supplied by the kit (Arbor Bioscience). Denatured and blocked libraries were combined with the pre-heated hybridisation mix containing the ASFV-specific biotinylated RNA-baits and incubated at the pre-set hybridisation temperature (65 °C) for 16 h for bait to target hybridisation. Subsequently, the hybridised bait–target mix was mixed with MyOne™ Streptavidin C1 beads (Thermo Fisher Scientific, Waltham, MA, USA). After immobilisation on a magnetic rack and washing using pre-heated washing buffer (65 °C), target sequences were eluted from the beads by adding 10 mM Tris-HCl, 0.05% TWEEN®-20 solution (pH 8.0–8.5) and incubating for 5 min at 95 °C. Enriched libraries were amplified using the AccuPrime™ Taq DNA Polymerase System (Thermo Fisher Scientific) according to the manufacturer’s instructions for 14 cycles prior to Illumina MiSeq sequencing [55]. Sequencing was performed using a MiSeq v3 600 cycle kit in 2 × 300 bp mode.

### 2.6. Data Analysis

Sequence data from the Illumina libraries lib02645 and lib02679 were mapped against the ASFV Georgia 2007/1 genome sequence (FR682468.1) using the Newbler 3.0 Software (Roche, Basel, Switzerland), and mapped reads were assembled using SPAdes 3.11.1 [53] in the mode of read error correction prior to assembly.

Basecalled fastq data from Nanopore sequencing were assembled in a hybrid assembly with the Illumina reads mapped to FR682468.1 (see above) using SPAdes v3.10.0. Contigs were used as a further reference for an iterative mapping approach using the generic mapper in Geneious v11.1.5 (Biomatters, Auckland, New Zealand). Regions with a coverage lower than 2.5 times standard deviations from the mean were excluded from consensus generation and further analysis. Nanopore fastq reads were used in a mapping approach in parallel using KMA [56], applying pre-sets optimised for Nanopore reads sensitive for indels. Coding sequences (CDS) were predicted with Glimmer3 [52] using FR682468.1 (ASFV Georgia 2007/1) for model generation. CDS were manually annotated and used to update INSDC entry FR682468.1, now available under FR6824681.2.

Single nucleotide variants (SNVs) were determined using the generic SNP finder of the Geneious software suite, applying parameters of a maximum *p*-value of 10^−5^ and filter for strand bias. The threshold for SNP identification was set at 10% and a minimum coverage of 100 was defined. Variants were checked manually for accuracy. Due to the low read quality at the genome ends, variants within the ITR regions were not included in the analysis. Read duplicates from all data used for SNV analysis and coverage estimation were removed using Samtools v. 1.9 [57] prior to analysis.

### 2.7. PCR and Sanger Sequencing

For conventional PCR, Phire Green Hot Start II PCR Master Mix (Invitrogen, Carlsbad CA USA) and the primers 457F CGGGCCAGACAAAATTGACC and 1223R AACAGGAAATACAAGGCGGC were used in a peqSTAR Thermocycler (VWR, Darmstadt, Germany) with the following cycling conditions: 30 s at 98 °C, 35 cycles each of 10 s at 98 °C, 30 s at 64 °C, 2 min at 72 °C and final elongation 10 min at 72 °C. Corresponding bands (751 bp) were purified from a 1.5% agarose gel with the QIAquick Gel Extraction Kit (Qiagen) and used for sequencing reactions (BigDye Terminator v3.1 Cycle Sequencing Kit, Applied Biosystems, Darmstadt, Germany). The reaction products were purified using NucleoSEQ columns (Macherey–Nagel) and sequenced on an ABI PRISM® 3100 Genetic Analyzer (Life Technologies, Darmstadt, Germany). Obtained sequences were assembled using the Geneious software, v11.1.5.

### 2.8. Data Availability

Whole-genome sequences are available from the International Nucleotide Sequence Database Collaboration (INSDC) databases under the study accession number PRJEB33279, and under the GenBank accession number FR682468.2.

## 3. Results

### 3.1. ASFV-Specific Target Enrichment Prior to Illumina Sequencing Provides High Amounts of Target Reads

For the resequencing of ASFV Georgia 2007/1, we compared a shotgun and a target enrichment approach. For shotgun sequencing, two libraries were prepared and sequenced; Table 2 summarises the key figures of these sequencing efforts. Library lib02645 was sequenced twice on the Illumina MiSeq platform. The first run resulted in 1,764,078 reads, from which 8438 reads (0.48%) were identified as ASFV-specific by mapping and resulted in a whole-genome sequence. However, the mean coverage of 13 was too low to assemble a reliable whole-genome sequence, especially in repeat and homopolymer regions. Therefore, in a second run, we sequenced 7,316,924 total reads, from which 36,670 reads were identified as ASFV-specific by mapping (0.5%), resulting in a whole-genome sequence with a mean coverage of 57 (Table 1) and leading to a better resolution.

Using target enrichment on the shotgun lib02679 prior to sequencing, we were able to increase the amount of ASFV reads identified by mapping to 44,862 from 60,174 total reads (66.78%), and whole-genome assembly was successful with a mean coverage of 57.2 (Table 1), also leading to a good resolution combined with less sequencing capacity.

However, the assembly of the resulting data was complicated by the extensive repeat regions in the ASFV genome as shown by the drop in coverage in the specific regions (Figure 2).

### 3.2. Hybrid Assembly of Nanopore and Illumina Data Provide an Improved ASFV Georgia 2007/1 Whole-Genome Sequence with Novel Information about the Genome Length

To overcome the limitations of short reads and to analyse the sequence of the inverted terminal repeat regions, long reads were generated by sequencing a library (lib02923) prepared from ASFV Georgia 2007/1 infected PBMCs using Nanopore sequencing. A total of 136,627 reads was generated with an estimated base calling accuracy of around 88.8%, and 1648 ASFV-specific Nanopore reads (1.20%) were identified by mapping and used for further analysis. The mean length of the used reads was 7881.4 nt, the median read length was 5517 nt, the N50 read length was 13,165 nt, with the longest read spanning was 43,474 nt (Appendix A). ASFV-specific Nanopore reads were used in a hybrid assembly with the ASFV-specific Illumina reads (lib02645, lib02679). The resulting contig was used as reference for further mapping and assembly, resulting in a new ASFV Georgia 2007/1 (FR682468.2) sequence with an improved overall coverage, particularly in repeat regions (Figure 2), and a length of 190,594 bp.

With a mean coverage of 67.5-fold, Nanopore reads were found to cover 100% of the ASFV Georgia 2007/1 reference sequence (Figure 2 and Appendix A) using a method not adapted to the Nanopore read characteristics. A mapping approach using only Nanopore reads and applying an algorithm with parameters adapted to the specific features of these data (KMA see methods) gained a consensus with only four differences (99.9% identical) to the reference sequence, with differences occurring in only one large homopolymer region (17xC in reference; 14xC in gained consensus), underlining the potential of this technique to support the assembly of large virus genomes and pointing to its constraints to dissolve homopolymers.

### 3.3. Sequence Alignment with the Improved ASFV Genome Reveals 71 Differences in Homopolymer Regions and Open Reading Frames

After gene predictions using Glimmer 3 in Geneious, the improved ASFV Georgia 2007/1 sequence was annotated on the basis of the available ASFV Georgia 2007/1 sequence. The annotation was manually checked and modified to accommodate changes in ORFs; e.g., split, fusion or truncations. Modified annotations include the ORFs (including small ORFs and ORFs within larger ORFs): MGF360 1L, G ACD 00070, G ACD 00160, 285L, MGF 110-10L, MGF110-14L, MGF 110-13L, G ACD 00290, MGF 300-2R, G ACD 01760, DP63R, MGF100-3L and MGF 360-19R (Appendix A).

Through the pairwise alignment of the original ASFV Georgia 2007/1 sequence (FR682468.1) with the improved ASFV Georgia 2007/1 sequence (FR682468.2), we identified 71 differences in the updated genome sequence leading to an overall sequence identity of 99.851%. From these differences, 58 are insertions or deletions (indels) in A/T homopolymer regions, seven are indels in G/C homopolymer regions, four are single nucleotide insertions and two are transversions. From these 71 differences, 22 are located in predicted and annotated open reading frames (ORF) causing frame shifts, while 49 do not affect ORFs. From the 22 frameshifts, eight lead to a shortened ORF, four to enlargement, three to the fusion of two ORFs, two to the split into two ORFs and five to the variation of codons in an ORF (Appendix A). Altogether, 13 changes affect ORFs previously shown to be expressed including the CP204L and KP177R ORFs coding for the structural proteins P30 (CP204L) and P22 (KP177R), A151R, C84L, QP383R, three members of the MGFs 110 and 360 and the small ORFs ASFV G ACD 00290, 350, 01760 (Appendix A). These differences to ASFV Georgia 2007/1 (FR682468.1) can also be identified in the published whole-genome sequences of ASFV Estonia 2014 [43], ASFV-POL/2015/Podlaskie [41] and ASFV Belgium 2018/1 [53] and are in agreement with differences to ASFV Georgia 2007/1 described in ASFV China [52], ASFV Georgia 2008 [45] and ASFV Poland [51].

As another important result, we were able to expand the 5’-ITR region of ASFV Georgia 2007/1 by 956 bp and the 3’-ITR region by 231 bp leading to a total length of 1378 bp of each ITR region, the annotation of the ORFs DP60R and ASFV G ACD 01990 at both ITR regions and a total genome length of 190,594 bp (Appendix A).

For validation of the in silico assemblies, the junction at the 5’ ITR was confirmed by PCR and Sanger sequencing.

### 3.4. Application of Target Enrichment Prior to Illumina Sequencing Enabled Whole-Genome Assembly of ASFV Moldova from Organ Samples Using the New Improved Sequence as Reference

Sequencing spleen samples from an ASFV infected pig from Moldova using a shotgun sequencing approach, we generated 8,232,518 reads, from which only 0.05% were ASFV-specific. However, the total amount of 4042 reads was not sufficient to generate a whole-genome sequence, neither by mapping nor by assembly (Table 1). By using the established target enrichment protocol on a shotgun library prior to sequencing, we were able to increase the amount of ASFV-specific reads to 690,206 (83.89%), from which 207,763 (25.0%) were unique. Mapping these data against the improved reference sequence (ASFV Georgia 2007/1—FR682468.2) and using SPAdes for the assembly, we were able to generate a whole-genome sequence of ASFV Moldova 2017/1 with a mean unique coverage of 317 (Table 1).

### 3.5. Variant Analysis Reveals Possible Single Nucleotide Polymorphisms

By mapping all data from ASFV Georgia 2007/1 libraries against the improved ASFV Georgia 2007/1 consensus sequence, we identified 49 locations with nucleotide variant frequencies of more than 10%. From these, 30 are located within 21 annotated ASFV-ORFs, leading to 26 amino acid substitutions, one truncation and one extension, while two do not change the corresponding amino acid. While the observed variant frequency is below 15% at 44 positions, we observed frequencies between 17.6% and 29.3% at positions 2963, 12,525 and 176,519. In addition, we observed frequencies in two poly G/C homopolymer regions of 40.8% and 47.7% (Appendix A).

For ASFV Moldova 2017/1, we identified 12 locations with nucleotide variant frequencies of more than 10%. From these, eight are located within seven annotated ORFs leading to frameshifts in six, a substitution and corresponding amino acid change in one and a silent mutation in another case (Appendix A).

### 3.6. Alignment of ASFV Moldova 2017/1 and ASFV Georgia 2007/1 Reveals Only Single Nucleotide Differences and a Tandem Repeat Variation

Comparing the ASFV Moldova 2017/1 genome with the improved ASFV Georgia 2007/1 sequence, we identified an overall sequence identity of 99.98% and 11 locations showing differences in single nucleotides (Appendix A). From these 11 locations, five were insertions or deletions in A/T homopolymer regions, three nucleotide transversions and three transitions. While four were located in non-coding regions, five are non-synonymous, leading to a codon change in annotated genes. From these, one leads to a stop codon integration causing the truncation of the MGF-110 1L ORF by 18 codons from the 5’-end (Appendix A) not affecting the predicted endoplasmic reticulum retention motif [58]. Furthermore, one leads to a frameshift, causing the variation of 23 and addition of five codons of the DP60R ORFs 5’-end as previously described in other whole-genome sequences from Eastern Europe [51] (Appendix A). In addition, we identified the integration of a 10 bp repeat in the previously described variable tandem repeat region between I73L and I329L [59] as well as differences in four non-coding poly G/C regions (Appendix A). Comparing these differences with all available Eastern European whole-genome sequences, only three of these differences were found exclusively in the genome sequence of ASFV Moldova 2017/1 (Appendix A).

### 3.7. Alignment and Phylogenetic Reconstruction of All Available ASFV Whole-Genome Sequences Shows a Very High Overall Nucleotide Sequence Identity of All Eurasian Strains

The alignment of all available ASFV whole-genome sequences revealed an overall nucleotide sequence identity of all Asian, Western and Eastern European whole genome sequences of over 99.9%. This high degree of identity is also shown by phylogenetic analysis; e.g., the resulting phylogenetic tree (Figure 3). While ASFV Moldova 2017/1 showed the highest nucleotide sequence identity to ASFV Belgium 2018/1 (99.99%), ASFV Georgia 2007/1 was most similar to ASFV Moldova 2017/1 and ASFV_Pol16_29413 _o23_ MG939586 (99.98%).

## 4. Discussion

In this study, we used a combined workflow including target enrichment, Illumina and Nanopore sequencing (Figure 4) for the resequencing of the ASFV Georgia 2007/1 isolate representing the first ASFV introduced into Eastern Europe in 2007.

Since the ASFV strains circulating in Eastern Europe and Asia are highly similar (Figure 3), molecular epidemiology analyses will rely on small changes and even single nucleotide variations. Therefore, the availability of high-quality sequences and the sharing of raw data as well as data on alignments and mappings is highly desirable to evaluate the quality of a sequence prior to analysis and search for a distinct pattern that could help to draw spatial and temporal conclusions.

With the use of ASFV target-specific enrichment prior to sequencing, we were able to increase the amount of target-specific reads allowing for whole-genome assembly while reducing the total number of reads considerably. As shown by our sequencing results, target-specific enrichment is especially useful when sequencing samples with a low virus to host ratio—e.g., organ tissue (Figure 4)—while samples with higher virus to host ratio—e.g., cell culture supernatant—do not necessarily call for enrichment (Figure 4).

However, since target-specific enrichment is performed using a bait set designed on known sequences, the enrichment and therefore the sequencing is limited to these. Although a hybridisation of baits to target sequences showing up to 90% identity can be expected [58], regions with extensive genetic variations or longer insertions might be missed. Therefore, bait sets should be designed carefully for the corresponding scientific question; e.g., specific sets for the whole-genome assembly of known viral genomes or sets including different genome variants for a broader enrichment. Furthermore, hybridisation parameters—e.g., temperature—can be optimised to include sequence variants [60]. In addition, the bias introduced by the amplification needed to generate the size of the library required for sequencing must be considered, especially when working on samples with a very low amount of target-specific sequences, and corresponding datasets need to be de-duplicated prior to analysis.

Since the ASFV genome includes extensive repeat regions, ITRs of unknown length and homopolymer regions with up to 17 C and 16 G, even data from target enrichment approaches have proven difficult for the de novo assembly of a high-quality reference sequence (Figure 2).

Through the employment of single molecule Nanopore sequencing (Figure 4), we were able to generate long viral reads of up to 43.5 kb. However, the low accuracy of the Nanopore reads (see above) hampers precise consensus sequence generation. While Nanopore reads reach an accuracy of around 90%, Illumina data sets usually have accuracies around 99.9%. Hence, the higher accuracy of the Illumina data sets can be used for better precision, whereas the Nanopore reads can be taken for correct assembly. Therefore, using these in a hybrid assembly, we were able to further improve the sequence quality, especially in the ITR regions. Although we were able to enlarge the ITR regions considerably, because of the decreasing sequence coverage at the ends, we cannot exclude the possibility that they are even longer. Due to their proposed involvement in ASFV replication involving head to head concatemers [61] (similar to poxviruses [62]), a complete sequence of the ITR regions including the terminal hairpin loops [63] would be especially interesting.

In our opinion, the presented workflows offer the best possible quality of ASFV sequences using the technology available to date. The sequences presented here are therefore of a reference nature and improve the analysis of additional ASFV sequences by far. Nonetheless, even with the combination of Nanopore and Illumina reads in a hybrid assembly, the sequencing of the poly G/C homopolymer regions has proven to be difficult, and we can neither completely exclude the existence of viral variants nor sequencing or bioinformatics artefacts, respectively. Therefore, it remains speculative if these homopolymer stretches might be involved in differential gene expression (maybe in the arthropod vector) or other unknown mechanisms, as might be suggested by the observed fusion or split of ORFs depending on the length of the homopolymer regions.

For all sequencing platforms, their specific sequencing limitations always need to be considered. For the 454 technology used for the generation of the original ASFV Georgia 2007/1 sequence (FR682468.1), limitations especially in sequencing homopolymer regions are well known [64]. Therefore, it is not surprising that we discovered numerous differences between the original and the improved sequence in A/T homopolymer regions, as also identified previously [41]. Since even the most recently developed sequencing technologies have their limitations, as shown by the evaluation of the IonTorrent and Illumina systems especially for homopolymer sequencing [64,65,66], selecting the adequate sequencing platform is crucial for each scientific question and should be done carefully. Furthermore, sequence information from the database should be evaluated thoroughly using the available information on the employed sequencing platform and data analysis prior to their use in experimental designs or bioinformatic analyses.

Using the ASFV-specific target enrichment prior to sequencing and the now available improved ASFV Georgia 2007/1 sequence as a reference, we could easily provide a reliable whole-genome sequence for ASFV Moldova 2017/1 from organ material without prior need for virus cultivation (Figure 4B). While the overall nucleotide sequence identity is more than 99.9% similar to the circulating ASFV strains from Eastern Europe and Asia, only a few single differences in nucleotides as well as a previously described tandem repeat insertion could be observed. However, the observed nucleotide differences, together with the available whole-genome sequence information, unfortunately do not allow for any further conclusions regarding phylogenetic or geographic relationships.

Although some of the single nucleotide changes in ASFV Georgia 2007/1 and ASFV Moldova 2017/1 reported in this study affect annotated ORFs, due to the lack of expression data for most of the ASFV ORFs and the lack of observations of virus attenuation from the field, no conclusion regarding their influence on virus replication or pathogenicity can be given. Therefore, transcriptome and proteome studies and infection experiments are needed to validate and update annotations, evaluate the influence of these changes on the viral phenotype and identify essential viral genes that might serve as targets for vaccine development (e.g., epitopes for T- or B-cells).

However, analysing SNVs in the viral genomes, we detected a clear difference in the number of SNVs. While we identified 49 SNVs in ASFV Georgia 2007/1, only 12 were detected in ASFV Moldova 2017/1. Since ASFV Georgia 2017/1 was passaged and sequenced from cell culture while ASFV Moldova 2017/1 was sequenced directly from organ material, it might be hypothesised that SNVs accumulate in cell culture-raised ASFV as was suggested for other viruses [67,68,69]. However, this observation might be influenced by the differences in coverage at the SNV sites, and although ASFV Moldova 2017/1 has a higher coverage, which should lead to a higher number of SNVs and a low sample size, further experiments with cell culture and field samples will be essential to confirm this hypothesis. In addition, more data on SNV in ASFV populations might be used in fine-tuning the intra-genotype phylogeny [70].

In conclusion, we used a workflow including target enrichment and Nanopore sequencing to provide an improved genomic sequence of ASFV Georgia 2007/1 (FR682468.2). Using this sequence as a reference, we subsequently generated the first whole-genome sequence from ASFV Moldova 2017/1, and showed that it is highly similar to the known circulating strains, showing only a few specific differences in single nucleotides from which only three seem to be specific for ASFV Moldova 2017/1.

We believe that our deep-sequencing workflow for ASFV and the provided reference sequences will be highly valuable to generate further high-quality ASFV whole-genome sequences and can serve as a basis for variant, transcriptome and proteome analyses. We are convinced that these sequences, along with coordinated efforts to improve data sharing and harmonised protocols, might pave the way for the identification of the new genetic markers that are desperately needed to understand virus evolution and trace routes of ASFV to eliminate the burden of this devastating animal disease.

## Figures and Tables

**Figure 1 viruses-11-00846-f001:**
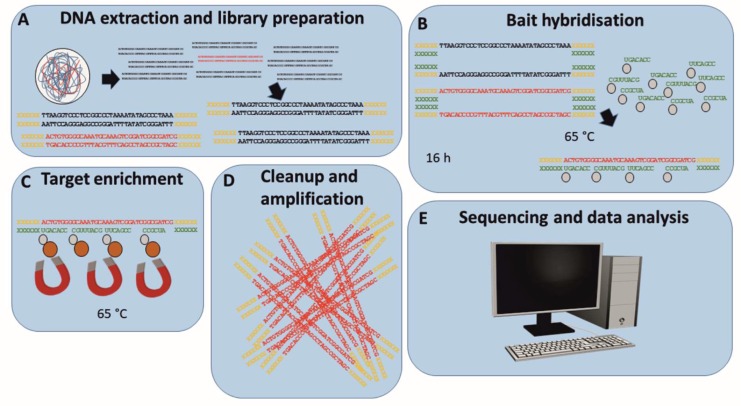
Workflow for the enrichment of ASFV-specific target sequences using MyBaits® (Arbor Bioscience) prior to Illumina MiSeq sequencing. Extracted DNA including ASFV target sequences (red) is fragmented and library is prepared by adding Illumina specific sequencing adapters (yellow) (**A**). After initial denaturation (95 °C), the library is cooled to hybridisation temperature (65 °C) and adapter specific blocking oligos (green) are added to prevent the re-hybridisation of adapter sequences during bait hybridisation (**B**). Biotinylated ASFV-specific RNA baits are added to the blocked library, and the reaction is incubated at hybridisation temperature for 16 h (**B**). Streptavidin-coated magnetic beads are added and bind to biotinylated RNA baits hybridised to ASFV target sequences (**C**). After magnetic separation and washing (**C**), ASFV target sequences are eluted from the baits and amplified (**D**). ASFV read-enriched libraries are sequenced on an Illumina MiSeq, and the resulting data are analysed in silico (**E**).

**Figure 2 viruses-11-00846-f002:**
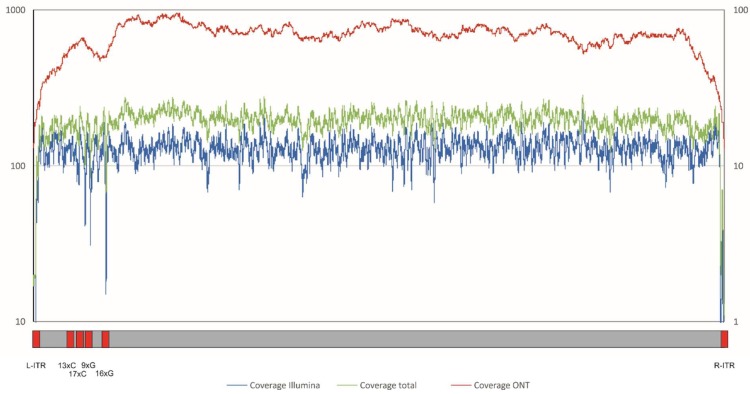
Coverage of ASFV Georgia 2007/1 with ASFV-specific reads from the different sequencing strategies. Blue line, coverage Illumina (left axis); red line, coverage Nanopore (right axis); green line, coverage total (Illumina + Nanopore) (left axis). Indicated below the figure are regions showing low coverage due to inverted terminal repeats (ITR) or G/C homopolymer stretches (red).

**Figure 3 viruses-11-00846-f003:**
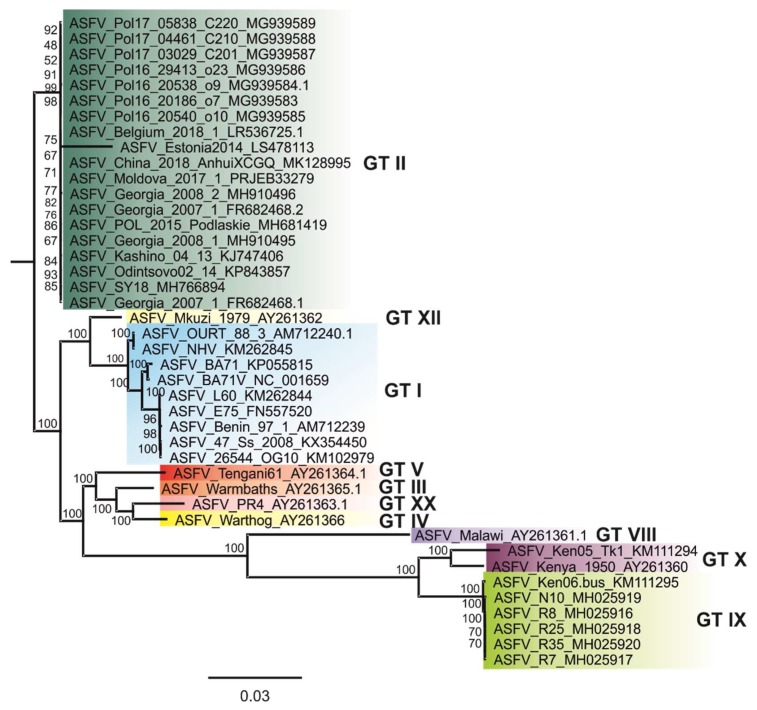
Phylogenetic tree showing all available ASFV whole-genome sequences. The maximum-likelihood (ML) tree was constructed using IQ-TREE v1.6.5 based on MAFFT v7.388 aligned ASFV whole-genome sequences. Standard model selection was used, resulting in the best-fit model K3Pu + F + R3 (three substitution types model + empirical base frequencies + FreeRate model with 3 categories). Statistical support of 10,000 ultrafast bootstraps is indicated at the nodes. Taxon names include, where available, ASFV-designation and INSDC accession number.

**Figure 4 viruses-11-00846-f004:**
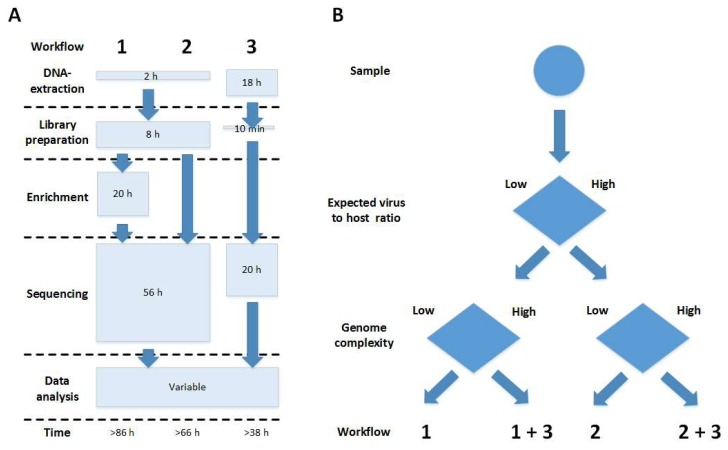
Comparison of sequencing workflows used in this study. (**A**) Overview of the possible strategies tested here for sequencing. (1) Illumina shotgun library preparation plus target enrichment prior to Illumina sequencing; (2) pure Illumina shotgun sequencing; (3) Nanopore sequencing. (**B**) Decision tree derived from the experiences in this study. Depending on the known or expected virus/host ratio and genome complexity, different workflows should be chosen. In this study, ASFV Georgia 2007/1 was treated as being a low virus/host ratio and high genome complexity sample and hence was sequenced according to all strategies (1/2/3), while ASFV Moldova 2017/1 had a low virus/host ratio but a suitable reference sequence was available, and hence target enrichment was sufficient, so only workflow (1) had to be applied.

**Table 1 viruses-11-00846-t001:** African swine fever virus (ASFV) whole-genome sequences available from public databases (status 11.03.2019).

Number	Accession Number	ASFV Isolate	Country of Origin	Submission Date	Collection Date	Host	P72 Genotype	WGS Publication	Method	Coverage
1	NC_001659.2	BA71V	Spain	1995	1967	Vero cells	I	[32]	Sanger sequencing	N/A
2	AY261360.1	Kenya 1950	Kenya	2003	1950	Domestic pig	X	N/A	N/A	N/A
3	AY261362.1	Mkuzi 1979	South Africa	2003	1979	Tick	XII	N/A	N/A	N/A
4	AY261365	Warmbaths	South Africa	2003	N/A	Tick	III / I	N/A	N/A	N/A
5	AY261363.1	Pretorisuskop/96/4	South Africa	2003	1996	Tick	XX / I	N/A	N/A	N/A
6	AY261361.1	Malawi Lil-20/1	Malawi	2003	1983	Tick	VIII	[34]	N/A	N/A
7	AY261366.1	Warthog	Namibia	2003	1980	Warthog	IV	N/A	N/A	N/A
8	AY261364.1	Tengani 62	Malawi	2003	1962	Domestic pig	V / I	N/A	N/A	N/A
9	AM712239.1	Benin 97/1	Benin	2007	1997	Domestic pig	I	[35]	Sanger sequencing	N/A
10	AM712240.1	OURT 88/3	Portugal	2007	1988	Domestic pig	I	[35]	Sanger sequencing	N/A
11	FN557520.1	E75	Spain	2009	1975	Domestic pig	I	[36]	Roche 454 GS FLX, Sanger sequencing	N/A
12	FR682468.1	Georgia 2007/1	Georgia	2010	2007	Domestic pig	II	[37]	Roche 454 GS FLX	N/A
13	KM102979.1	26544/OG10	Italy (Sardinia)	2014	2010	Domestic pig	I	[46]	Illumina HiScanSQ, Sanger sequencing	20
14	KJ747406.1	Kashino 04/13	Russia	2014	2013	Wild boar	II	N/A	Sanger sequencing	N/A
15	KM111295.1	Ken06.Bus	Kenya	2014	2006	Domestic pig	X	[39]	Illumina HiSeq	N/A
16	KM262844.1	L60	Portugal	2014	1960	Domestic pig	I	[38]	Amplicon sequencing on Roche 454 GS FLX, Sanger sequencing	N/A
17	KP055815.1	BA71	Spain	2014	1971	Domestic pig	I	[44]	Sanger sequencing	N/A
18	KM262845.1	NHV	Spain	2014	1968	Domestic pig	I	[38]	Amplicon sequencing on Roche 454 GS FLX, Sanger sequencing	N/A
19	KM111294.1	Ken05/Tk1	Kenya	2015	2005	Tick	IX	[39]	Illumina HiSeq	N/A
20	KP843857.1	Odintsovo_02/14	Russia	2015	2014	Wild boar	II	N/A	Roche 454 GS FLX	N/A
21	LP643842.1	Patent WO2015091322	N/A	2015	N/A	N/A	N/A	N/A	N/A	N/A
22	KX354450.1	47/Ss/2008	Italy (Sardinia)	2016	2008	Domestic pig	I	[40]	Illumina MiSeq; PacBio	N/A
23	MG939585.1	Pol16_20540_o10	Poland	2018	2016/2017	Sus scrofa	II	[51]	Illumina MiSeq	20-40
24	MG939587.1	Pol17_03029_C201	Poland	2018	2016/2017	Sus scrofa	II	[51]	Illumina MiSeq	20-40
25	MG939583.1	Pol16_20186_o7	Poland	2018	2016/2017	Sus scrofa	II	[51]	Illumina MiSeq	20-40
26	MG939588.1	Pol17_04461_C210	Poland	2018	2016/2017	Sus scrofa	II	[51]	Illumina MiSeq	20-40
27	MG939584.1	Pol16_20538_o9	Poland	2018	2016/2017	Sus scrofa	II	[51]	Illumina MiSeq	20-40
28	MG939586.1	Pol16_29413_o23	Poland	2018	2016/2017	Sus scrofa	II	[51]	Illumina MiSeq	20-40
29	MG939589.1	Pol17_05838_C220	Poland	2018	2016/2017	Sus scrofa	II	[51]	Illumina MiSeq	20-40
30	MH681419.1	ASFV/POL/2015/Podlaskie	Poland	2018	2015	Wild boar	II	[41]	Illumina MiSeq	103
31	MH766894.1	ASFV-SY18	China	2018	2018	Domestic pig	II	N/A	N/A	N/A
32	MH025918.1	R25	Uganda	2018	2015	Domestic pig	IX	[42]	Illumina NextSeq 500	869
33	MH025920.1	R35	Uganda	2018	2015	Domestic pig	IX	[42]	Illumina NextSeq 500	1487
34	MH025917.1	R7	Uganda	2018	2015	Domestic pig	IX	[42]	Illumina NextSeq 500	439
35	MH025916.1	R8	Uganda	2018	2015	Domestic pig	IX	[42]	Illumina NextSeq 500	309
36	MH025919.1	N10	Uganda	2018	2015	Domestic pig	IX	[42]	Illumina NextSeq 500	23
37	LS478113.1	Estonia 2014	Estonia	2018	2014	Domestic pig	II	[43]	Illumina MiSeq	100
38	MH910495.1	Georgia 2008/1	Georgia	2018	2008	Domestic pig	II	[45]	Illumina MiSeq	8.5
39	MH910496.1	Georgia 2008/2	Georgia	2018	2008	Domestic pig	II	[45]	Illumina MiSeq	118
40	MK128995.1	China/2018/AnhuiXCGQ	China	2019	2018	Domestic pig	II	[52]	BGISEQ-500	271
41	LR536725.1	Belgium 2018/1	Belgium	2019	2018	Wild Boar	II	[53]	Illumina MiSeq	292

N/A: data not available; WGS: whole-genome sequence.

**Table 2 viruses-11-00846-t002:** Comparison of shotgun and target enrichment sequencing approaches for the generation of ASFV whole-genome sequences.

ASFV	Sample Type	Library Number	Sequencing Mode	Total Reads	Total ASFV Reads	% ASFV Reads	Mean Coverage
Georgia 2007/1	Cell culture supernatant	lib02645	shotgun	1,764,078	8309 (8150)	0.47 (0.46)	12.7 (12.5)
		lib02645	shotgun	7,317,744	36,268 (33,454)	0.5 (0.46)	56.5 (52.1)
		lib02679	myBaits	67,174	44,862 (40,234)	66.78 (59.89)	57.2 (51.7)
Moldova 2017/1	Spleen	lib02487	myBaits	829,408	690,206 (207,763)	83.89 (25.0)	1055 (317)
		lib02577	shotgun	8,232,518	4042 (3986)	0.05 (0.048)	N/A

() Number/percentage of unique reads and coverage of the corresponding ASFV whole-genome sequence are given in brackets.

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
