# Peer review of "A Deep-Sequencing Workflow for the Fast and Efficient Generation of High-Quality African Swine Fever Virus Whole-Genome Sequences"

_viruses, 2019, doi:10.3390/v11090846_

Round 1

Reviewer 1 Report

Abstract

Semantic remarks

Line 27: comma before which

Line 27f: only few … only for a few, maybe replace with only for some

Introduction

Minor remarks:

Line 56: Whereby only warthogs have been shown to be directly part of the sylvatic cycle, it’s important to note that all wild suids, including bush pig and red river hog and giant forest hogs can be asymptomatic carriers of ASFV.

Line 74: Based on recent developments add Myanmar to the list of affected Asian countries

Line 80: Homologous recombination in large DNA viruses has been found to be a very important cause for the mutation rate. This is the main reason why whole genome sequencing for ASFV is of utmost importance. I’d suggest to reference to publications on Vaccinia that studied this in more detail, e.g. Qin & Evans, JVI, 2014.

Line 89 ff: I am not sure whether the general virologist is going to comprehend the shortcomings of paired-end reads, assembly, and the consequences to the reference genome. It might be worth bulking this section out to highlight yet again the importance of the work on hand.

Semantic remarks:

Line 59: comma before which

Line 69: reintroduced… by introduction, maybe remove by introduction and change into Georgia to in Georgia

Materials and Methods

Line 113: I presume PBMs are porcine primary peripheral blood monocytes – is this supposed to be PBM or PBMC? Abbreviation not introduced

Line 115: Same thing if PBMs and PBMCs are the same thing the reference and abbreviation should be added to/ introduced in line 113.

 Line 116fff: Just a remark, whilst kits can be good good, in our hands traditional Phenol/Chloroform produces higher quality (longer fragments) and larger quantities, which, especially for long-read DNA for PacBio and Nanopore sequencing is advantageous.

Results

ASFV enrichment

Line 229: Why was a shotgun library used for Nanopore sequencing? If anything shearing should be avoided to get as close to whole genome sequences as possible. At the same time the read lengths do not quite correspond to what would be expected from a shotgun library, can the authors elaborate on this here or in the M&Ms, and definitely expand the library generation protocol in the M&Ms – e.g. what shearing protocols were used etc.?

Elaborating on this here; we are able to easily get long reads, and the whole ASFV genome read should be achievable. The authors have made good decisions in using the rapid kit for this since, even though it has a tendency to shear DNA, contrast to other of the Nanopore kits it has less pipetting and vortexing steps, avoiding shearing.

The coverage in figure S1 depicts nicely the difficult areas. In such a technical paper, this information that should be in the main manuscript, so I’d suggest to add the figure but also read lengths in the Nanopore sequencing as a figure. It would also be interesting to see where the long Nanopore reads align to in the sequence. Also the drop in sequencing in high repeat regions should be highlighted. Maybe a schematic depiction of the genome could be added underneath the alignment / coverage.

Hybrid assembly of Nanopore and Illumina data

A line on the accuracy of base calling with Nanopore would be interesting here.

Application of enrichment on organ samples

Can the authors clarify whether they once again used a shotgun approach / fragmented the DNA before target enrichment. Yet again if they did this is a shame. If there are worries about large complex formation there are various compounds that can aid relax the DNA and in our hands don’t interfere with the Nanopore sequencing, such as boric acid and DMSO.

Discussion

The authors are correct in highlighting that care must be placed on bait selection. Here non-baited Illumina sequencing could help with comparing baits used with sequence in the sample to optimise baits for the sample on hand if necessary. Also, if the authors would not fragment the DNA before baiting, there is a higher chance of obtaining variants in other regions. This optimisation of the protocol should be discussed – or conversely, if the authors tried that approach and it didn’t work then this should be highlighted.

In the respective results section the authors talk about the longest Nanopore sequence from virus supernatant being 43.5kb, yet in the respective section (lines 349fff) the authors refer to the longest read being 71kb. As mentioned above, this information and read length distribution should be added to a figure, also to assess whether the 71kb or 43.5kb are a rare exception with low coverage or not. This also allows to assess the shotgun method used for library preparation in the first place.

Lastly, the authors should discuss accuracy of the MinION Nanopore and why or if Illumina sequencing is necessary to corroborate the sequence accuracy. They could also speculate whether multiplexing or the use of a Flongle could be possible cost savings for the sequencing of ASFV.

Author Response

Abstract

Semantic remarks

Line 27: comma before which

Line 27f: only few … only for a few, maybe replace with only for some

Changed according to the reviewers suggestions

Introduction

Minor remarks:

Line 56: Whereby only warthogs have been shown to be directly part of the sylvatic cycle, it’s important to note that all wild suids, including bush pig and red river hog and giant forest hogs can be asymptomatic carriers of ASFV.

Line 74: Based on recent developments add Myanmar to the list of affected Asian countries

--> Changed according to the reviewers suggestions

Line 80: Homologous recombination in large DNA viruses has been found to be a very important cause for the mutation rate. This is the main reason why whole genome sequencing for ASFV is of utmost importance. I’d suggest to reference to publications on Vaccinia that studied this in more detail, e.g. Qin & Evans, JVI, 2014.

--> In general, we agree with the reviewers comment. However, for ASFV, HR between different viral isolates has never been reported. In addition, especially in Eastern Europe where only highly similar ASFV strains from genotype II are circulating, HR between these seems possible, but it remains unclear if larger genetic changes can be expected. However, the passage has been modified to clarify that.

Line 89 ff: I am not sure whether the general virologist is going to comprehend the shortcomings of paired-end reads, assembly, and the consequences to the reference genome. It might be worth bulking this section out to highlight yet again the importance of the work on hand.

--> The passage was modified according to the reviewers suggestions

Semantic remarks:

Line 59: comma before which

Line 69: reintroduced… by introduction, maybe remove by introduction and change into Georgia to in Georgia

--> Changed according to the reviewers suggestions

Materials and Methods

Line 113: I presume PBMs are porcine primary peripheral blood monocytes – is this supposed to be PBM or PBMC? Abbreviation not introduced

--> Clarified according to the reviewers suggestions

Line 115: Same thing if PBMs and PBMCs are the same thing the reference and abbreviation should be added to/ introduced in line 113.

--> Clarified according to the reviewers suggestions

 Line 116fff: Just a remark, whilst kits can be good good, in our hands traditional Phenol/Chloroform produces higher quality (longer fragments) and larger quantities, which, especially for long-read DNA for PacBio and Nanopore sequencing is advantageous.

Results

ASFV enrichment

Line 229: Why was a shotgun library used for Nanopore sequencing? If anything shearing should be avoided to get as close to whole genome sequences as possible. At the same time the read lengths do not quite correspond to what would be expected from a shotgun library, can the authors elaborate on this here or in the M&Ms, and definitely expand the library generation protocol in the M&Ms – e.g. what shearing protocols were used etc.?

Elaborating on this here; we are able to easily get long reads, and the whole ASFV genome read should be achievable. The authors have made good decisions in using the rapid kit for this since, even though it has a tendency to shear DNA, contrast to other of the Nanopore kits it has less pipetting and vortexing steps, avoiding shearing.

--> For nanopore sequencing, DNA extracted manually by chlorophorm/phenol extraction as stated in lines 128ff. Prior to Nanopore sequencing, the DNA was not additionally sheared. The term “shotgun” should just imply that this library was not enriched for ASFV-sequences. However, to clarify, we remove the term “shotgun”.

The coverage in figure S1 depicts nicely the difficult areas. In such a technical paper, this information that should be in the main manuscript, so I’d suggest to add the figure but also read lengths in the Nanopore sequencing as a figure. It would also be interesting to see where the long Nanopore reads align to in the sequence. Also the drop in sequencing in high repeat regions should be highlighted. Maybe a schematic depiction of the genome could be added underneath the alignment / coverage.

--> Figure 1 was moved to the main part. A figure showing length distribution of mapped nanopore reads and the alignment of the 80 longest of them in respect to the ASFV genome was added to the supplement. More data about the length distribution of mapped nanopore reads was also added to the text.

Hybrid assembly of Nanopore and Illumina data

A line on the accuracy of base calling with Nanopore would be interesting here.

--> For the Guppy v2.1.3, the base calling accuracy is known to be around 91%.(Oxford Nanopore Technologies). The base calling accuracy in this experiment was slightly lower (88.8 %). This was added to the text.

Application of enrichment on organ samples

Can the authors clarify whether they once again used a shotgun approach / fragmented the DNA before target enrichment. Yet again if they did this is a shame. If there are worries about large complex formation there are various compounds that can aid relax the DNA and in our hands don’t interfere with the Nanopore sequencing, such as boric acid and DMSO.

--> Since we used only Illumina MiSeq sequencing for ASFV target specific enrichment, the library had to be constructed according to Illumina specifications. Following the myBaits protocol, library preparation (including fragmentation) has to be done prior to target enrichment to enable direct amplification (using the adapters) after enrichment thereby avoiding loss of the low molarity unique library molecules. Since we sequenced at 300 bp paired end mode, the library was shared and size-selected aiming at a peak at 550 bp fragment length. However, we added the information for clarification.

 Discussion

The authors are correct in highlighting that care must be placed on bait selection. Here non-baited Illumina sequencing could help with comparing baits used with sequence in the sample to optimise baits for the sample on hand if necessary. Also, if the authors would not fragment the DNA before baiting, there is a higher chance of obtaining variants in other regions. This optimisation of the protocol should be discussed – or conversely, if the authors tried that approach and it didn’t work then this should be highlighted.

--> As stated above, library preparation (including fragmentation) for Illumina miSeq sequencing prior to enrichment was set by the mybaits protocol. However, further experiments will include the enrichment for nanopore sequencing using baits therefore overcoming this obstacle.

In the respective results section the authors talk about the longest Nanopore sequence from virus supernatant being 43.5kb, yet in the respective section (lines 349fff) the authors refer to the longest read being 71kb. As mentioned above, this information and read length distribution should be added to a figure, also to assess whether the 71kb or 43.5kb are a rare exception with low coverage or not. This also allows to assess the shotgun method used for library preparation in the first place.

--> Figure S1 was moved to the main manuscript and information about read length and reads length distribution added to the text

Lastly, the authors should discuss accuracy of the MinION Nanopore and why or if Illumina sequencing is necessary to corroborate the sequence accuracy. They could also speculate whether multiplexing or the use of a Flongle could be possible cost savings for the sequencing of ASFV.

--> The points raised by the reviewer concerning the accuracy of Nanopore reads have been integrated in the discussion.

--> Multiplexing of samples is available and could lead to cost reduction if the samples are of high enough quality and concentration but we have not tried that yet for ASFV. Unfortunately, at the time of sequencing, Flongles were not commercially available and have yet to be tested for ASFV. However, if the amount of pores in a Flongle proves to be sufficient for a full ASFV genome, their usage could lead to further savings. The possibility to combine multiplexing and Flongles still needs further testing in order to assess the efficiency and in line the opportunity for cost reduction. Therefore, we would not integrate this in the discussion at that time.

Reviewer 2 Report

Summary: This manuscript demonstrates a new way of generating the sequence of very large DNA virus genomes by taking advantage of new nanopore technology that does not require prior replication of the virus in cell culture. Using this technology in tandem with Illumina NGS, the authors generated a "fuller"-length and corrected sequence of an early isolate of ASFV from Georgia as well as a genome sequence for an isolate from Moldova. The results of this analysis should provide the field with accurate reference sequences for future studies.

The manuscript is well written and the target audience will easily be able to follow the procedures; figure 1 is especially useful for those outside this audience.  My issues with this manuscript are minor.

Critique:

1) My major concern is with the results/discussion of the Moldova sequence variations, specifically in the annotated genes MGF 110-1L and DP60R.  Please describe the truncation of MGF 110-1L and the "altered" ORF of DP60R (is this a frameshift/fusion?) relative to loss of known functional domains.  A recent manuscript describes the FLS of a Poland isolate that has a frameshift in DP60R - are you seeing the same type of change?  This manuscript is by Mazur-Panasiuk, et al, Scientific Reports, Mar 2019.

2) The variations that are seen here represent genetic drift during in vivo spread of ASFV across Europe over a roughly ten-year period.  Knowing what specific changes are occurring can apprise the scientist about genes that are possibly redundant or have domains that are not absolutely necessary for replication in swine.  But are there also "hot-spots" for variation that change somewhat "randomly" during replication that have little significance on virulence?  It seems that the significance of these changes may not be very high and may improperly skew the importance of future observations.  For example, the expansion of the a sequence in HSV, while important for packaging, varies widely among isolates and doesn't seem to have an absolute size requirement.  Similar to the poly(C) tract in FMDV.  Please comment in the discussion about your opinion of the significance of these homopolymer tracts in ASFV, possibly in discussion lines 365/366.

3) Regarding potential hotspots for variation, many cell culture adapted ASF viruses have been generated and sequenced.  Can you determine if some of the homopolymer changes are often seen in the sequences of these viruses?  This might provide information about changes that are "less" important for replication and virulence.  If these comparisons exist, please comment in the discussion - this could go a long way at determining the significance of variations in the homopolymer regions.

Author Response

Critique:

1. My major concern is with the results/discussion of the Moldova sequence variations, specifically in the annotated genes MGF 110-1L and DP60R.  Please describe the truncation of MGF 110-1L and the "altered" ORF of DP60R (is this a frameshift/fusion?) relative to loss of known functional domains.  A recent manuscript describes the FLS of a Poland isolate that has a frameshift in DP60R - are you seeing the same type of change?  This manuscript is by Mazur-Panasiuk, et al, Scientific Reports, Mar 2019.

--> The description of the changes of DP60R and MGF110-1L ORFs and cross-reference to Supplementary Table S2 (where the changes are described) has been added.

--> The changes in DP60R are identical to the differences observed in the polish isolates. For MGF110-1L, the predicted endoplasmic reticulum retention motif is unchanged but the 3’-region is truncated by 18 codons. This has been also cross-referenced and added to the manuscript.

2. The variations that are seen here represent genetic drift during in vivo spread of ASFV across Europe over a roughly ten-year period.  Knowing what specific changes are occurring can apprise the scientist about genes that are possibly redundant or have domains that are not absolutely necessary for replication in swine.  But are there also "hot-spots" for variation that change somewhat "randomly" during replication that have little significance on virulence?  It seems that the significance of these changes may not be very high and may improperly skew the importance of future observations.  For example, the expansion of the a sequence in HSV, while important for packaging, varies widely among isolates and doesn't seem to have an absolute size requirement.  Similar to the poly(C) tract in FMDV.  Please comment in the discussion about your opinion of the significance of these homopolymer tracts in ASFV, possibly in discussion lines 365/366.

--> A statement concerning a possible influence of the poly G/C homopolymer stretches has been added to the discussion.

-->Concerning hotspots of variation, comparing some of our high quality whole genome sequences of ASFV and the relating data (e.g. mappings etc) genetic variation has been observed mainly in the 5’- and 3’- variable regions while the core genome is only rarely affected. However, a clear “hot-spot” in terms of specific genes regularly affected could not be identified yet.

3. Regarding potential hotspots for variation, many cell culture adapted ASF viruses have been generated and sequenced.  Can you determine if some of the homopolymer changes are often seen in the sequences of these viruses?  This might provide information about changes that are "less" important for replication and virulence.  If these comparisons exist, please comment in the discussion - this could go a long way at determining the significance of variations in the homopolymer regions.

--> Since the resolution/reliability of the existing NGS-platforms in poly G/C regions is not sufficient, unfortunately we cannot use these for analysis yet. Basically, we see different lengths of the poly G/C-regions in most of the sequences we compared with no special emphasis on cell-culture/non-cell culture strains.